# Lower Rate of Water Absorption of an Oral Rehydration Solution with Partially Hydrolyzed Guar Gum in Conscious Rats

**DOI:** 10.3390/nu14204231

**Published:** 2022-10-11

**Authors:** Toru Takahashi, Miki Tokunaga, Tsutomu Okubo, Makoto Ozeki, Mahendra P. Kapoor, Zenta Yasukawa

**Affiliations:** 1Department of Nutrition, Faculty of Nutrition, Kanazawa Gakuin University, Kanazawa City 920-1392, Ishikawa, Japan; 2Nutrition Division, Taiyo Kagaku Co., Ltd., Yokkaichi 510-0844, Mie, Japan

**Keywords:** water absorption, rats, Bayesian network, small intestine, PHGG

## Abstract

The purpose of the present study was to elucidate the rate of water absorption of an oral rehydration solution containing partially hydrolyzed guar gum (PHGG) in the small intestine, which is the main site of water absorption when water is drunk. Seven-week-old male SD rats were employed. We prepared four types of an aqueous solution, i.e., containing mineral and sugar, containing PHGG, containing mineral and sugar and PHGG, or containing no additives. After 24 h of food and 4 h of water deprivation, the aqueous solutions were infused into the stomach of conscious rats on their hands using a syringe with a stomach sonde. We sampled the stomach and the small intestine with contents 8 min after the infusions. Causal effects were calculated using a Bayesian network. PHGG increased the residual amount of water in the gastro-intestine, which depends negatively on the absorption of water in the small intestine/the flow rate to the small intestine. The absorption of water in the small intestine depended positively on the flow rate to the small intestine, which depended negatively on the free water in the solutions. PHGG decreased water absorption in the small intestine by decreasing the free water in the aqueous solutions.

## 1. Introduction

Dehydration is a common problem among the elderly and athletes. It is associated with decrements in the performance of an activity, cognition, and mood [1,2,3]. Appropriate treatment for dehydration depends on assessing water and electrolyte deficits and slowly correcting those deficits [1].

A higher rate of water absorption often decreases the osmotic pressure of the blood, which can stimulate water excretion via the renin-angiotensin-aldosterone system [4]. Hence, a lower rate of water absorption might induce longer retention of water in the body. Some athletes in sports such as football and marathon restrict the duration and frequency of drinking water during physical activities. The elderly show less thirst [5]. It is advantageous for athletes and the elderly to increase the retention of water in their bodies.

In the past, studies on water absorption have examined the perfusion of the small intestine under anesthesia [6]. However, the perfusion and/or anesthesia might affect water absorption in the small intestine. The present study evaluated water absorption in the small intestine in conscious rats without the perfusion experiment.

Water absorption might be controlled by the differentiation of the water potential between the intestinal lumen and tissues [7]. The water potential can be calculated by the sum of solute and pressure potential in the body of mammals [8]. Factors affecting the water potential are levels of sugar and cations such as Na, K, Mg, free water content, the apparent diffusion coefficient (ADC) of water, and osmotic pressure, as noted in previous studies [7,8,9]. Hence, the present study deals with these factors. Such conditions for aqueous solutions could be satisfied with an oral rehydration solution. The free water content—one of the factors of water potential—of the aqueous solutions was controlled by the addition of fibers [10]. Our previous study confirmed that partially hydrolyzed guar gum (PHGG) decreased free water because of its bonding qualities [11]. The purpose of this study was to elucidate the rate of water absorption of an oral rehydration solution with PHGG in the small intestine, which is the main site of water absorption when water is drunk. We also discuss manners of sugar absorption involving PHGG additives and Na absorption. The modulation of cations and PHGG may be a potential strategy for the management of dehydration in the elderly and athletes.

## 2. Materials and Methods

### 2.1. Animals

Twenty-four 7-week-old male SD rats were purchased from Japan SLC, Inc. (Shizuoka, Japan) and were housed individually in stainless-steel cages with wire-mesh bottoms under a 12:12 h light: dark cycle at 24 °C. The rats had free access to a standard commercial diet (CE-2; Japan SLC) and water for 7 days before the start of the experiment. The mean body weight of the rats at the time of the ingestion of the sample at 8 weeks of age was 323 ± 12 g. The rats were maintained following the guidelines for animal experimentation of the Japanese Association for Laboratory Animal Science at the Faculty of Human-Environment Science at Fukuoka Women’s University.

### 2.2. Aqueous Solution for Water Absorption

We prepared four types of aqueous solutions, i.e., containing mineral and sugar, containing PHGG, containing mineral and sugar and PHGG, or containing no additives (control); see Table 1. The composition of the aqueous solution containing minerals, sugar and PHGG accorded with that of Sunfiber^®^ Water (Taiyo Kagaku Co. Ltd., Yokkaichi city, Japan). All solutions included polyethylene glycol (PEG), which was an internal standard for measurements of water absorption [12]. The recovery rate of PEG in the stomach and small intestine also showed the validity of the estimation of water absorption in the small intestine.

### 2.3. Infusion of Aqueous Solutions

The aqueous solutions were prepared in a beaker at 40 °C on a heat stirrer for ca. 5 min and then loaded into a 10 mL syringe with a stomach sonde. Before and after infusions, the weights of the syringes with stomach sondes were measured to confirm the volume of the infusion. After 24 h of food and 4 h of water deprivations, the aqueous solutions were infused into the stomachs of conscious rats using a syringe with a stomach sonde. No restraining tool was used. The mean infusion volume was 0.038 mL/kg BW with SE 0.002. The ratio of stomach volume to rat body weight cannot be compared with that of humans. The tissue of the proximal part of the stomach of the rat accords to the tissue of the esophagus, called the forestomach [13]. A compartment of the stomach of the rat consists of the esophagus and the stomach [13]. Accordingly, the stomach volume of the rat relative to its body weight is much higher than that of humans. There were no differences in the infusion volume between groups.

### 2.4. Sampling

We applied four bulldog clamps on the proximal and distal ends of the stomach and small intestine at 8 min after the infusion of the aqueous solution under deep anesthesia with diethyl ether. The time was measured using a stopwatch with an alarm. We sampled the stomach and the small intestine, each with 2 clamps, by cutting with a scalpel immediately after setting clumps. Within 60 s of the sampling, the digestate in the gastric and small intestinal lumen was collected and its weight was measured. After sampling, the rats were killed by transection of the jugular vein while still under anesthesia.

### 2.5. Water Activity and Free Water Content in Aqueous Solutions before Infusion

Water activities and the contents of free water were determined using the apparent diffusion coefficient (ADC) of water and T2 in the aqueous solutions utilizing a 1.5 T MRI (TOSHIBA) at the Suzuka University of Medical Science.

### 2.6. Meaurements of Cations, PEG, Sugar and Osmotic Pressure

Cations such as Na^+^, K^+^, and Mg^2+^ were determined using an ion chromatograph (761 Compact IC, Metrohm). PEG was measured using nephelometry, as described in a previous study [12]. Levels of sugar in the aqueous solutions and digest were measured using the enzyme method (Glu-CII, Wako, Singapore) and a spectrophotometer (Shimazu). The osmotic pressures of the aqueous solutions and digest were measured using the freezing point depressing method (OM6020, Arkley, Inc., Kyoto, Japan).

#### Calculation of Water and Cations

Water secretion in the stomach was calculated by summing the water secretion at every minute (from 1 to 8 min), as follows:
$$\mathrm{Water}\;\mathrm{secretion}\;\mathrm{in}\;\mathrm{the}\;\mathrm{stomach}\;=\;\begin{array}{cc} \sum_{k=1}^{8}\{(\mathrm{weight}\;\mathrm{of} & \mathrm{contetnts}\cdot \mathrm{content}\;\mathrm{of}\;\mathrm{PEG}\;\mathrm{in}\;\mathrm{gastric}\;\mathrm{contetnts}/2)/8\}\cdot k\\ & \cdot \{\mathrm{weight}\;\mathrm{of}\;\mathrm{gastric}\;\mathrm{contetnts}\cdot (1\\ & -\mathrm{content}\;\mathrm{of}\;\mathrm{PEG}\;\mathrm{in}\;\mathrm{gastric}\;\mathrm{contetnts}/2)/8\}/(k\\ & \cdot (\mathrm{weight}\;\mathrm{of}\;\mathrm{gastric}\;\mathrm{contetnts}\cdot (1\\ & -\mathrm{content}\;\mathrm{of}\;\mathrm{PEG}\;\mathrm{in}\;\mathrm{gastric}\;\mathrm{contetnts}/2)/8)+(\mathrm{volume}\;\mathrm{of}\;\mathrm{injection}\\ & -k\cdot (\mathrm{volume}\;\mathrm{of}\;\mathrm{injection}-\mathrm{weight}\;\mathrm{of}\;\mathrm{gastric}\;\mathrm{contetnts}\\ & \cdot \mathrm{content}\;\mathrm{of}\;\mathrm{PEG}\;\mathrm{in}\;\mathrm{gastric}\;\mathrm{contetnts}/2)/8)), \end{array}$$

Flow rate to the small intestine = volume of injection + volume of secretion in the stomach.

Absorption of water in the small intestine = flow rate to the small intestine − volume of the digest in the small intestine.

Secretion of cations in the stomach = amount of cations in the small intestine + cations present into the small intestine – cations present in the aqueous solution.

The amount of cations present into the small intestine = $\sqrt{\mathrm{content}\;\mathrm{of}\;\mathrm{cations}\;\mathrm{in}\;\mathrm{the}\;\mathrm{stomach}\cdot \mathrm{content}\;\mathrm{of}\;\mathrm{cations}\;\mathrm{in}\;\mathrm{the}\;\mathrm{aqeuos}\;\mathrm{solution}}$. Flow rate to the small intestine.

Cations absorbed in the small intestine = cations present in the small intestine – cations present in the small intestine.

The amount of cations present into the small intestine was calculated by multiplying geometric mean of the content of cations by the flow rate to the small intestine.

### 2.7. Statistics

Causal effects were calculated using a Bayesian network [14]. This approach can indicate causal relationships using Bayes’ theorem between variables without dependence on graph theory; thus, the Bayesian network shows causation between variables by using arrows in a graph. A Bayesian network is a directed acyclic graph that is composed of a set of variables {X1, X2, …, XN} and a set of directed edges between the variables [14]. A variable has several possible states, for example, true and false. Bayesian networks are widely used in probabilistic knowledge representation and reasoning. In Bayesian networks, the joint probability distribution function of all nodes can be calculated as follows:$$P\;(X1,\;X2,\;\ldots ,\;\mathrm{XN})=\overset{N}{\underset{i=1}{\prod }}P(X_{i}|Pa_{i}),$$
where, Pa_i_ is the set of random variables whose corresponding nodes are parent nodes of X_i_.

A Bayesian network contains two elements: structure and parameters. In our study, each arc begins at a parent node and ends at a child node. Pa (X) represents the parent nodes of node X. X1 is the root node because it has no input arcs. Root nodes have prior probabilities. Each child node has conditional probabilities based on the combination of states of its parent nodes. Black circles represent discrete variables, and white circles represent ordinal variables.

The correlation coefficients between measurement items were analyzed using Pearson’s correlation coefficients. The effects of the PHGG additive were analyzed using two-way ANOVA.

Statistical analyses were performed using R version 3.5.2 (The R Project for Statistical Computing, Vienna, Austria) with R Studio Version 1.0.143 (R Studio, Inc.) and SYSTAT version 13.2 (Hulinks Inc., Tokyo, Japan). The results were expressed as means with standard errors.

We analyzed the effects on variables of sugar and mineral additives and of PHGG in the aqueous solutions using two-way ANOVA, with subsequent Tukey multiple comparisons if interactions were significant. Variables for two-way ANOVA were elected by the results of the Bayesian network. A probability of less than 0.05 was regarded as significant.

## 3. Results

The recovery rate of PEG in the stomach and small intestine was 94% in the present study.

### 3.1. Causal Effects between Variables

Figure 1 shows the causal effects between the variables measured, as calculated using Bayesian network theory. Arrowheads and lines indicate effects and causes, respectively.

The Bayesian network indicated that the factor affecting the absorption of water in the small intestine was the flow rate to the small intestine (Figure 1). The factor affecting the flow rate to the small intestine was the water secretion in the stomach and free water in the aqueous solution. The factors affecting the free water in the aqueous solution were the sugar, cation, and PHGG additive levels in the aqueous solution. The factors affecting the absorption of sugar in the small intestine were cation and sugar levels, osmotic pressure, and ADC in the aqueous solution, as well as the absorption of Na in the small intestine. The factor affecting ADC in the aqueous solution was the free water in the aqueous solution. One of the factors affecting the free water in the aqueous solution was the PHGG additive (Figure 1).

### 3.2. Associations between Variables in Figure 1

For a detailed analysis and better visualization, we integrated the results of the Bayesian network (Figure 1) with the results of the correlation coefficients and ANOVA into the schema in Figure 2. Accordingly, Figure 2 shows causal relationships with quantitative relationships. Correlations between numerical variables and differences on all arrows are shown in Figure 2. The presence of a “+” on the arrows in the panel in Figure 2 indicate positive correlations while “–” indicates negative correlations with *p* < 0.05. Associations between discrete and numerical variables were analyzed using ANOVA. The words “Control” and “PHGG” on the arrows indicate that control was higher or lower than PHGG, respectively, with *p* < 0.05.

### 3.3. Effects of PHGG and Solutes in the Variables Involving Water Absorptions

We selected the variables involving water absorption in the results of the Bayesian network shown in Figure 2. Figure 3 shows panels of the free water in the aqueous solution, the flow rate to the small intestine, the absorption of water in the small intestine, the water absorption in the small intestine divided by the flow rate to the small intestine, and the residual amount of water in the gastro-intestine, with ANOVA tables. The present study employed two-way ANOVA because two factors for the free water of the aqueous solutions existed in Figure 2, i.e., effect of a mixture of sugars and cations in the aqueous solutions and the effect of PHGG. The latter was significant in the flow rate to the small intestine, the water absorption in the small intestine divided by the flow rate to the small intestine, and the water absorption in the small intestine flow rate to the small intestine (*p* < 0.05, Figure 3). Interaction between PHGG and a mixture of sugars and cations was significant (*p* = 0.001, Figure 3). The absorption of sugars in PHGG with the addition of sugar and minerals was lower than that of no PHGG with sugar and minerals (Tukey multiple comparison, *p* < 0.05, Figure 3).

## 4. Discussion

The present study discussed the rate of water absorption of aqueous solutions containing PHGG in the small intestine. According to our findings, the recovery rate of PEG from the stomach and small intestine was 94%, which might validate the estimation of water absorption in the small intestine.

We employed a duration of 8 min to absorb the aqueous solutions. In a preliminary experiment, a duration of 15 min showed a plateau of water absorption from all aqueous solutions and no differences in water absorption among the aqueous solutions. Accordingly, all aqueous solutions were completely absorbed after 15 min in the small intestine when the aqueous solutions were injected into the stomach. This suggests that the results of the present study, employing the duration of 8 min, will not show the amount of water absorption. Hence, the present study was able to discuss the rate of water absorption in the small intestine.

### 4.1. Water Absorption and PHGG

We summarized the results of water and sugar absorption with the results of a Bayesian network with quantitative relationships (Figure 2) and the results of two-way ANOVA of variables involving water and sugar absorptions (Figure 3) in Figure 4. That figure shows that PHGG increased the residual amount of water in the gastro-intestinal tract, which depended negatively on the absorption of water in the small intestine/the flow rate to the small intestine. The absorption of water in the small intestine depended positively on the flow rate to the small intestine, which depended negatively on the free water in the aqueous solutions (Figure 1 and Figure 4). Furthermore, the PHGG additive decreased the free water in the aqueous solutions (Figure 1 and Figure 4). PHGG likely decreased water absorption in the small intestine by decreasing the free water in the aqueous solutions. Accordingly, an oral rehydration solution with PHGG could slow the rate of water absorption.

As mentioned above, it is advantageous for athletes and the elderly to increase the retention of water in the body. A slower rate of water absorption in the small intestine might increase the retention of water, reducing the risk of dehydration. Accordingly, an oral rehydration solution with PHGG may be useful to avoid dehydration among athletes and the elderly.

### 4.2. Absorption of Sugar with PHGG

PHGG decreased the rate of glucose absorption in rat small intestine and reduced increments of postprandial plasma glucose, as noted in our previous study [11]. The present study also confirmed that PHGG decreased the rate of absorption of sugars in the small intestine (Figure 1 and Figure 4). PHGG simultaneously decreased free water and the ADC of water in the aqueous solution, which affected the absorption of sugars (Figure 1 and Figure 4). PHGG decreased the absorption of sugars when comparing the aqueous solutions containing sugar and minerals and those with sugar, minerals, and PHGG (Tukey multiple comparison, *p* < 0.05, Figure 3).

The ADC of water and/or free water content might be among the modulators of glucose absorption in the small intestine, which, in turn, reduces postprandial blood glucose.

### 4.3. Contradictions between Intake of NaCl and Absorption of NaCl Involved in the Absorption of Sugar

The absorption of sugar might be affected by the absorption of Na in the small intestine (Figure 4). The absorption of molecular amounts of sugars and Na were 301 and 3.28 mole, respectively, in the aqueous solution with sugar and mineral additives. In cases of lower absorption of Na compared with glucose, the absorption of former might stimulate the absorption of latter.

In a standard Japanese man in his 60 s with a 2600 kcal intake, an activity level of “2” [15] and a NaCl intake of 10.7 g [16], the molecular amount of sugar and Na in the daily diet are ca. 2.0 mole and 0.18 mole, respectively. Thus, the ratio of the molecular amount of sugar to that of Na to be absorbed in the small intestine might be ca. 10 times higher. Accordingly, such a subject can absorb sugar 10 times more rapidly than Na on a molecular basis. In our study, conscious rats absorbed sugar 100 times more rapidly than Na (Figure 3). On the other hand, SGLT 1, which is the main transporter of glucose in the epithelium of the small intestine, needs two moles of Na^+^ for the absorption of one mole of glucose [17], suggesting that the ratio of sugar to Na must be 0.5. A Japanese man in his 60 s has to absorb 4 mole of Na in the small intestine to co-transport sugar if SGLT 1 needs two units of Na to absorb one unit of sugar. The 4 mole of Na accords to 234 g of NaCl per day. However, the WHO recommends a NaCl intake of less than 5 g (approximately 2 g of Na) per person per day for the prevention of cardiovascular diseases. Contradictions exist between intake of NaCl and transport across the epithelium of NaCl involving the co-transportation of sugar. Na secretion from saliva and gastric fluid might compensate for Na supply in the gastro-intestinal lumen for the co-transportation of Na and sugar. However, the maximum Na secretion in rat stomach was 170 mg in the present study. Na secretion into the gastro-intestinal lumen cannot reach 234 g of Na supplied to the small intestinal lumen in humans. Furthermore, the secretion of Na in the stomach did not affect any factors in the present study (Figure 1). Rather, a secretion mechanism of Na in the stomach to reduce the concentration of Na in the small intestinal lumen might exist as a means to improve the absorption of sugar.

### 4.4. Limitations

The present study could not distinguish the effects of cations from those of sugars. This may have obscured the results regarding the absorption of sugars. However, the design of the present study revealed details about the mechanism of the absorption of water.

The present study could not compare the effects of cations with those of sugar on sugar absorption because of the existence of confounding factors, namely, the correlation between the ADC of water in the aqueous solution and the absorption of sugars in the small intestine (Figure 3 and Figure 4), which might have given rise to misleading results. The design of the present study might leave room for further improvements. However, the main purpose of this study was to determine the effects of PHGG on water absorption using oral rehydration solutions. In this regard, it achieved its main purpose to some extent.

The present study employed the single dose model to measure water absorption in the small intestine. The single dose model was established to eliminate confounders in glucose and water absorption studies, as mentioned in Section 2.2 [7,11]. We did not focus on the differences between the single dose and perfusion models. This might be seen as a limitation relative to other papers.

## 5. Conclusions

The present study employed the combination of an internal standard, conscious rats, mathematical calculations, and a Bayesian network to elucidate the effects of PHGG in oral rehydration solution on water absorption. Our findings revealed some aspects regarding the mechanisms of water and glucose absorption in the small intestine. PHGG in oral rehydration solution decreased the rate of water absorption in the small intestine by decreasing free water, which is considered to reduce dehydration among athletes and the elderly. The rate of sugar absorption was also slowed in the presence of PHGG by decreasing the apparent diffusion coefficient of water, which, in turn, may reduce increments of postprandial blood glucose, as shown in a previous study. Controlling free water and the apparent diffusion coefficient of water with water-soluble fibers such as PHGG could provide an alternative strategy for the prevention of dehydration and hyperglycemia.

## 6. Patents

T. Takahashi1, M. Morita, M. Tokunaga, M. Tokunaga, T. Mitzuya and Zenta Yasukawa (P6584777, Japan) 2015.

## Figures and Tables

**Figure 1 nutrients-14-04231-f001:**
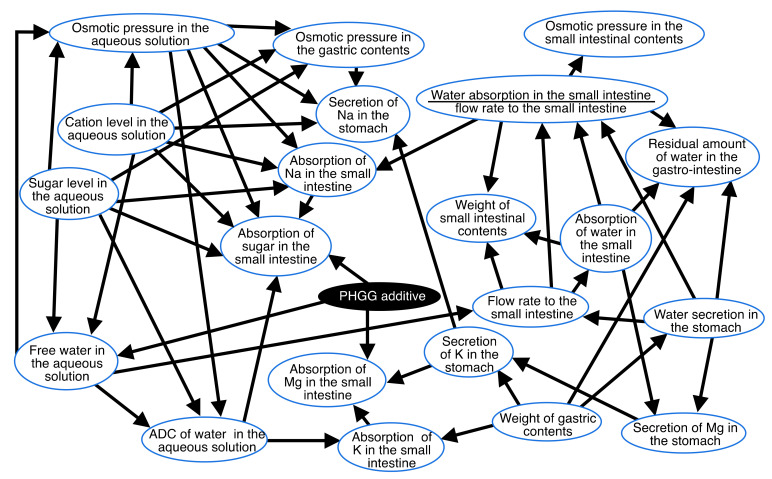
Causal effects between the variables, as determined using the Bayesian network. Arrowheads and lines indicate effects and causes, respectively.

**Figure 2 nutrients-14-04231-f002:**
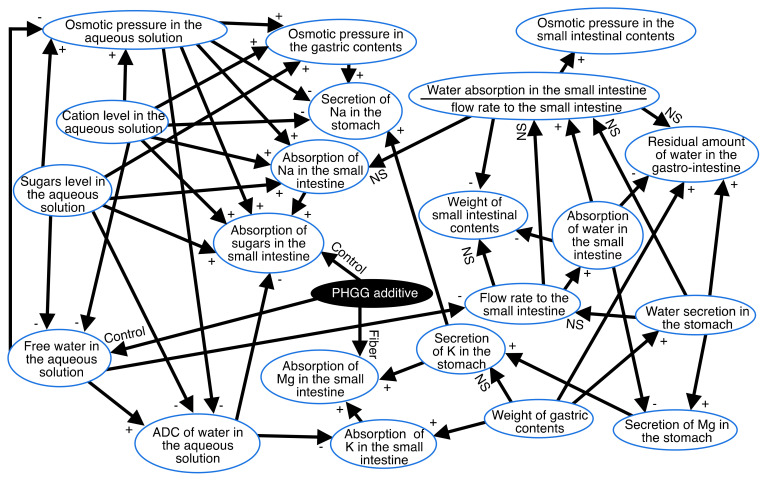
Causal and quantitative relationships: a combination of the results of the Bayesian network and Pearson’s correlation and two-way ANOVA. The presence of a “+” on arrows indicates positive correlations while “–” indicates negative correlations with *p* < 0.05. Associations between discrete and numerical variables were analyzed using ANOVA. The words “Control” and “PHGG” on the arrows indicate that the control was higher or lower than PHGG, respectively, with *p* < 0.05.

**Figure 3 nutrients-14-04231-f003:**
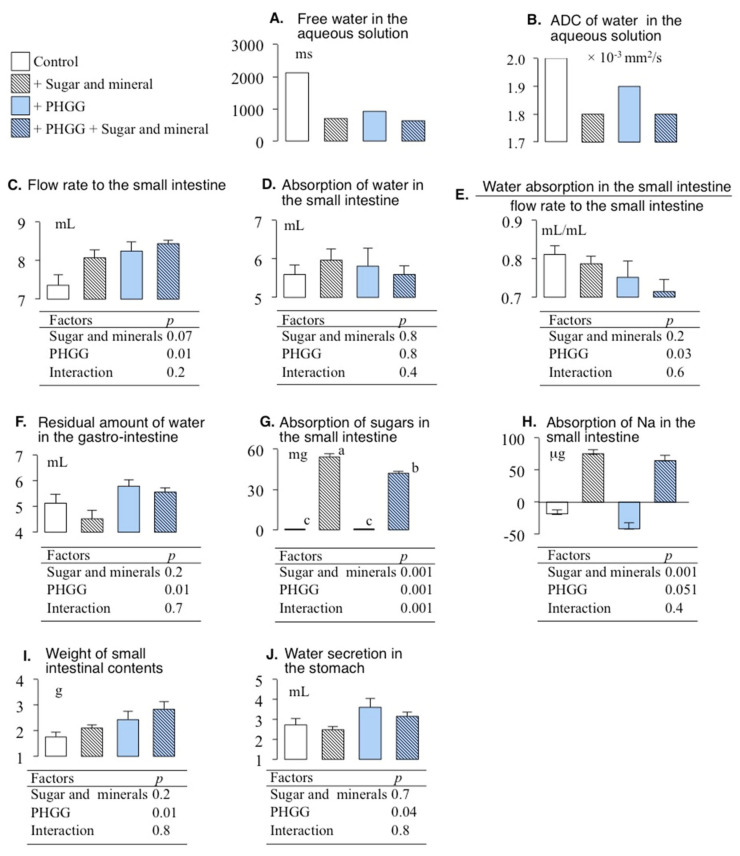
ANOVA tables and the (**A**) free water in the aqueous solution, (**B**) ADC of water in the aqueous solution, (**C**) flow rate to the small intestine, (**D**) the absorption of water in the small intestine, (**E**) the water absorption in the small intestine divided by the flow rate to the small intestine, (**F**) the residual amount of water in the gastro-intestinal tract, (**G**) the absorption of sugars in the small intestine, and (**H**) absorption of Na in the small intestine, (**I**) the weight of the contents of the small intestine, and (**J**) water secretion in the stomach with the effect of the mixture of sugars and cations in the aqueous solution, and the effect of PHGG.

**Figure 4 nutrients-14-04231-f004:**
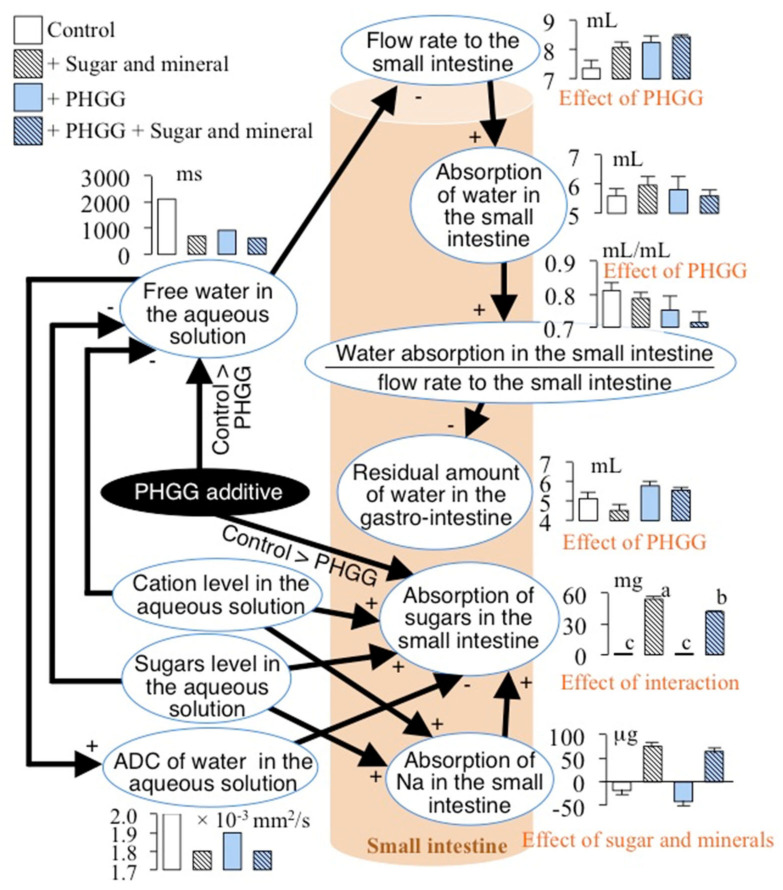
Summary of water and sugar absorption in the small intestine. The letters of the effects in red in the panel shows significant at *p* < 0.05 analyzed by two-way ANOVA (Figure 3).

**Table 1 nutrients-14-04231-t001:** Composition of aqueous solutions containing sugars and minerals, PHGG, sugars and minerals and PHGG.

Composition (g/L)	Control	Addition of Sugar and Minerals	Addition of PHGG	Addition of PHGG and Sugar and Minerals
NaCl	-	1.30	-	1.30
KCl	-	1.15	-	1.15
Ca_3_(PO_4_)_2_	-	0.280	-	0.280
Na_3_C_6_H_5_O_7_	-	1.30	-	1.30
Citric acid	-	2.00	-	2.00
Ascorbic acid	-	0.500	-	0.500
MgSO_4_	-	0.172	-	0.172
Sucralose	-	0.0280	-	0.0280
Acesulfame potassium	-	0.120	-	0.120
Sucrose	-	10.0	-	10.0
Glucose	-	12.8	-	12.8
PHGG	-	-	10.0	10.0
PEG	5.00	5.00	5.00	5.00
Flavor of Yuzu	-	0.400	-	0.400

PHGG: Partially hydrolyzed guar gum, PEG: polyethylene glycol, Yuzu: Citrus junos.

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
