# Peer review of "Lower Rate of Water Absorption of an Oral Rehydration Solution with Partially Hydrolyzed Guar Gum in Conscious Rats"

_nutrients, 2022, doi:10.3390/nu14204231_

Round 1

Reviewer 1 Report

The manuscript investigated the effects of an oral dehydration solution with PHGG on water absorption in conscious rats through the combination of an internal standard, Wistar rats, mathematical calculations, and a Bayesian network.

Modifications are needed.

Ø  Just one gavage of PHGG is not sufficient.

Ø  Single sex experiment of Wistar rats is incomplete to support the conclusion “Modulation of cations and PHGG may be a potential strategy for the management of dehydration in the elderly and athletes.”

Ø  In section 2.2, the unit of infusion should be mL/kg·bw.

Author Response

Response to the editor comments.

  1. Just one gavage of PHGG is not sufficient.

Page 3 and 11-12, We added sentences about the single dose model in 2.2 and limitation.

  1. Single sex experiment of Wistar rats is incomplete to support the conclusion “Modulation of cations and PHGG may be a potential strategy for the management of dehydration in the elderly and athletes.”

Page 3 and 11-12, We added sentences about limitations of sex in limitation.

  1. In section 2.2, the unit of infusion should be mL/kg·bw.

Page 3, We revised the unit of infusion as suggested.

Reviewer 2 Report

In this manuscript, the authors attempt to elucidate the water absorption rate of an oral rehydration solution with partially hydrolyzed guar gum (PHGG) in the small intestine. They investigate the conscious rat's water absorption in the small intestine without the perfusion experiment. They claim that PHGG would decrease water absorption by decreasing the free water in the aqueous solutions. The following points should be clarified.

1  The authors should compare the results of the methods used in this paper with that of perfusion experiments.

2  Please add a rationale and explanation for the formula used by the authors.

3  Page 2; Table 2 Table 1

4  In Figure 4, the bar fill pattern differs between the bar description and the bar chart.

Author Response

Response to the reviewer comments.

  1. The authors should compare the results of the methods used in this paper with that of perfusion experiments.

 Page 3 and 11-12, We added sentences about the single dose model in 2.2 and limitation.

2  Please add a rationale and explanation for the formula used by the authors.

  Page 4 and 5, We added sentences about explanation for the formulain 2.2.

3  Page 2; Table 2 →Table 1

 Page 3, We revised the table number as suggested.

4  In Figure 4, the bar fill pattern differs between the bar description and the bar chart.

Figure 4, We changed description as suggested.

Round 2

Reviewer 1 Report

The single injection dose of 0.038 mL/kg BW in Wistar rat was much lower than the reported doses in many literatures. What is the basis of your choice?

Author Response

Response to Reviewer 1.

 The single injection dose of 0.038 mL/kg BW in Wistar rat was much lower than the reported doses in many literatures. What is the basis of your choice?

1.  Yes, the single injection dose of 0.038 mL/kg BW was higher than that of paper focusing on effects of solutes.  However, effects of single injection dose without anesthesia on the water absorption had not been reported in the rat.  We can't compare the  injection dose between our study and others in water absorption.

2. Yes, we chose the injection dose. The  single injection dose of 0.038 mL/kg BW didn't induce vomiting. We can't detect any changes of behaviors of the rats.  On the results of observation of behavior of the rats injecting 0.038 mL/kg water in the preliminary study, we chose the volume.

Reviewer 2 Report

The authors addressed the points which I noted.

Author Response

We confirmed the comment of the Reviewer 2.